# Amharic Audio Data Search Engine using Text-Based Spoken Term Detection with Models

## Abstract

The generation of audio files from various sources, including the internet and social media, has increased significantly in the rapidly expanding digital landscape. It is difficult to efficiently access specific spoken words from this vast collection of Amharic audio data. To address this, we propose a novel method that combines Text-Based Spoken Term Detection (STD) with models. Our methodology includes speech segmentation with pydub, the development of an ASR model, and the implementation of keyword-based STD. The ASR model successfully transcribes audio files, allowing meaningful keywords to be extracted for more accurate and frequent search queries. An analysis of 37 audio files reveals that the sentence error rate (SER) is 91.7 percent (33 of 36 sentences have errors) and the word error rate (WER) is 98.3 percent (285 of 290 words have errors). It improved search accuracy and efficiency for specific spoken terms, significantly improving search capabilities for users of Amharic multimedia resources. However, the study emphasizes the need for a larger dataset to improve transcription capabilities and reduce errors, with the potential to revolutionize Amharic audio search engines and empower users in accessing precise information from Amharic audio data, ultimately transforming how we interact with and use Amharic audio resources.

Keywords: ASR (Automatic Speech Recognition), STD (Spoken Term Detection), WER (Word Error Rate), SER (Search Error Rate)

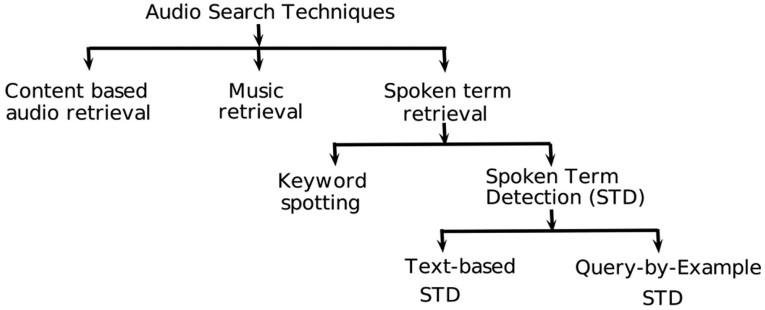

Figure 1: Techniques of searching

## 1 Introduction

In the rapidly expanding digital landscape, a vast number of audio files are being generated from diverse sources such as the internet and social media, contributed by individuals and organizations. Among these valuable resources are audio files in Amharic, the official language of Ethiopia [1][2]. However, the challenge lies in efficiently searching for specific spoken words within this extensive collection of Amharic audio files.

Traditional audio players lack the essential functionality for word-based audio search, leaving users grappling to extract desired information from the stored Amharic audio data[4]. To overcome this obstacle, the development of specialized audio search engines becomes imperative, capable of matching and retrieving relevant documents exclusively from the database of stored Amharic language audio files. By enabling such advanced audio search capabilities, users can save time and improve their ability to access precise information tailored to their needs.

The primary goal of this study is to address this pressing need by exploring effective methods for searching particular words in audio files recorded in Amharic. The findings hold the potential to revolutionize information retrieval through the creation of cutting-edge search engines specifically designed for Amharic multimedia. These advancements will empower individuals, groups, and organizations relying on Amharic audio data, enabling them to seamlessly access pertinent information and identify specific passages within the audio files, ultimately transforming the way we interact with and harness the potential of Amharic audio data.

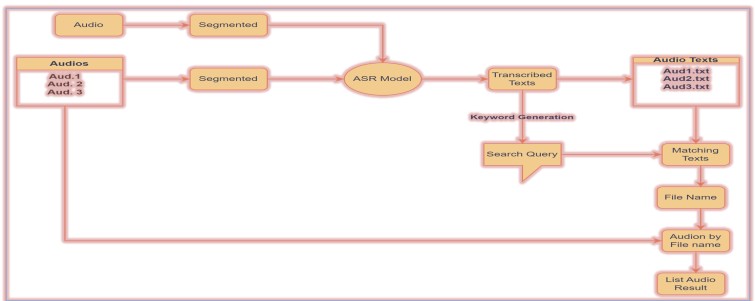

Figure 2: General Flow chart

## 2 Implementation Detail

Speech Segmentation is the first step in our methodology, where we divide a long speech signal into smaller segments. This segmentation process, facilitated by the Python package pydub, divides the audio file into frames based on silence, making it more manageable for further analysis.

The creation of an ASR (Automatic Speech Recognition) model follows speech segmentation. We train the segmented audio chunks to their corresponding text transcriptions using the CMU Sphinx, an open-source ASR development toolkit [3]. We utilize the Sphinx4 speech recognition library in conjunction with Python to build and interface our ASR system [4][5], resulting in a seamless alignment of various components.

To find specific spoken speech within an audio file using Amharic text as a keyword, we investigate various searching methods, including STD (Spoken Term Detection), QBE STD (Query By Example STD), and keyword detection. Given our research's primary goal of identifying specific spoken words in an audio file using Amharic text, we choose keyword-based Spoken Term Detection (STD) as our preferred searching method. This strategy enables efficient and precise location of the desired speech fragments within the audio file.

Our research's central methodology revolves around the integration of various tools and techniques, including CMU Sphinx, Sphinx4, Python, and pydub. This integration process empowers us to efficiently and precisely search for specific spoken terms in the Amharic audio files, contributing to the development of advanced search engines tailored to Amharic multimedia.

| Aspect | Description |
|---|---|
| Audio Segmentation | Segmentation area 12-minute audio file into smaller chunks using PyDub. |
| ASR Model Creation | The ASR model is created with a language model comprising 2737 sentences and 6231 words, along with n-gram language models (1-gram, bigram, and trigram). The acoustic model is trained with 100 audio samples, containing 247 phones, 1882 words, and a phonetic dictionary. |
| Evaluation | The trained ASR model is evaluated on 37 audio files. The evaluation results indicate a Sentence Error Rate (SER) of 91.7% (33 out of 36 sentences have errors) and a Word Error Rate (WER) of 98.3% (285 out of 290 words have errors), F-1 score,precision and recall results |
| Matching Audio Transcriptions | The code matches audio files with their corresponding transcriptions and conducts keyword-based searches. |
| Recommendation for Improvement | The text suggests that the ASR model's performance is limited due to a small dataset and recommends using a larger dataset for training to enhance accuracy. |

Figure 3: Table

# 3   Experiment

As illustrated in Figure 2, The Searching Audio, which is transcribed using an ASR model to create a keyword, and the List of Audio to be Searched, which is also transcribed using the same ASR model, are the two audio files that are used as input in this process. The transcriptions of the List of Audio files are used to create an index, which makes efficient querying possible. The generated keyword is then used as a search query to locate and playback audio files from the List of Audio that match the specified term. The retrieved audio files are then printed or displayed as output. In this process, two audio files,"The Searching Audio" and the "List of Audio to be Searched," are transcribed using an ASR model to generate keywords.The transcriptions of the "List of Audio" are used to create an index, facilitating efficient searches. The generated keyword acts as a search query to identify and retrieve audio files from the "List of Audio" that match the specified term. These retrieved audio files are then presented as the output. This approach streamlines the process of finding and accessing relevant audio content, as depicted in workflow the performance shows all the WER, CER, Precision, recall, and F-1 scores.

## 4 Results and conclusion

pydub is used to segment a 12-minute input audio wave file into smaller chunks during the training phase. Following that, an ASR model is created using a language model with 2737 sentences and 6231 words, as well as n-gram language models (1-gram: 6231, bigram: 14731, trigram: 16079). The acoustic model was trained using 100 audio samples, which lasted 12 minutes and included 247 phones and 1882 words. A phonetic dictionary with 373 phone numbers is also used.The trained ASR model is evaluated on 37 different wave files. The results show that the sentence error rate (SER) is 91.7% (33 of 36 sentences have errors) and the word error rate (WER) is 98.3% (285 of 290 words have errors). These evaluation metrics are critical in determining the ASR model's performance and accuracy for the given task. As illustrated in Figure 2, The list of audios is correctly matched by the complete transcription of the audios and the search using a most frequent keyword-base as the search query.The small dataset used for training is one of the ASR model's main limitations. To improve transcription capabilities and reduce errors, use a large dataset

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
