# OpenReview forum: "Amharic Audio Data Search Engine using Text-Based Spoken Term Detection with Models"
_wikimedia.it/Wikidata_and_Research/2025/Conference — Submitted to WD&R_

### Official Review · ~Camillo_Carlo_Pellizzari_di_San_Girolamo1 · 2024-12-30
**Seemingly no pertinence with Wikibase and/or Wikidata**

**Originality:** 3
**Impact:** 2
**Confidence:** 3

**Review:**

The abstract does not show any clear connection with Wikidata and/or Wikibase, which are the themes of the conference; it might be hypothesized that the ASR model could be useful for the Wikibase instance Lingua Libre (https://lingualibre.org/) and the lexemes in Wikidata.

**Compliance:**

2

**Notes:**

-

**Scientific Quality:**

3

---

### Official Review · ~Iolanda_Pensa1 · 2025-01-12
**Lingua Libre**

**Originality:** 3
**Impact:** 2
**Confidence:** 3

**Review:**

The proposal doesn't seem pertinent for the conference Wikidata and research.

I can see some potential links with the Wiki project https://lingualibre.org, which adds recordings made by the communities to Wikimedia Commons and links them to Wikipedia and Wikidata (Lingua Libre is really a great project already well-connected with the Wikimedia projects). Maybe it could be interesting for the author to explore those synergies for the future.

**Compliance:**

1

**Scientific Quality:**

3

---

### Decision · Program_Chairs · 2025-02-05

**Decision:**

Reject

**Comment:**

Dear Zemenfes Hailemariam Gebremedhin,
thank you for your submission. We regret to inform you that your proposal was not selected because not centered around Wikidata.
Thank you very much for submitting a proposal to the conference and our best wishes for your work.
Regards,
The scientific committee of the conference Wikidata and Research